# The Cotton Dust-Related Allergic Asthma: Prevalence and Associated Factors among Textile Workers in Nam Dinh Province, Vietnam

**DOI:** 10.3390/ijerph18189813

**Published:** 2021-09-17

**Authors:** Tran Thi Thuy Ha, Bui My Hanh, Nguyen Van Son, Hoang Thị Giang, Nguyen Thanh Hai, Vu Minh Thuc, Pham Minh Khue

**Affiliations:** 1Faculty of Public Health, Haiphong University of Medicine and Pharmacy, Haiphong 04212, Vietnam; tttha@hpmu.edu.vn (T.T.T.H.); htgiang@hpmu.edu.vn (H.T.G.); nthanhhai@hpmu.edu.vn (N.T.H.); 2Department of Tuberculosis and Lung Disease, Hanoi Medical University, Hanoi 11521, Vietnam; buimyhanh@hmu.edu.vn; 3Department of Occupational Diseases, National Institute of Occupational and Environmental Health, Hanoi 11611, Vietnam; nguyenvansonbnn@yahoo.com; 4Department of Research and Training, Tam Anh Hospital, Hanoi 11813, Vietnam; vuminhthuc2010@gmail.com

**Keywords:** allergic asthma, cotton dust, textile workers, Nam Dinh, Vietnam

## Abstract

Objective: To determine the prevalence of cotton dust-related allergic asthma and associated factors among textile workers in Nam Dinh province, Vietnam. Methods: A cross-sectional study was performed with 1082 workers in two textile garment companies using the asthma diagnostic criteria of the GINA (Global Initiative for Asthma) 2016 guidelines. Results: Among study participants, 11.9% had suspected asthma symptoms, 7.4% were diagnosed with asthma, and 4.3% (3.6% in men and 4.5% in women) were diagnosed with cotton dust-related allergic asthma. Overweight, seniority more than 10 years, history of asthma, allergic rhinitis, family history of allergy, and exposure to cotton dust from more than one hour per day in the working environment were found to be important predictors of cotton dust-related allergic asthma among textile workers. Conclusions: Textile workers in two companies in Nam Dinh, Vietnam had a high prevalence of dust-related allergic asthma compared to estimates from the general population. There is a need to design appropriate measures of prevention, screening, and care for dust-related asthma in the textile industry. Further evaluation with better exposure assessment is necessary.

## 1. Introduction

Asthma is a common non-communicable chronic respiratory disease and occurs in all countries regardless of the level of development. The definition of asthma still has many different opinions and statements. According to WHO, asthma is characterized by recurrent attacks of breathlessness and wheezing, which vary in severity and frequency from person to person [1]. According to Gina’s report based on various data sources, there are approximately 358 million people with asthma in the world (with global prevalence ranging from 1 to 22% of the population) [2]. In addition, 26.2 million disability-adjusted life years (DALYs) are estimated to be lost due to asthma, representing 1.1% of the total burden of disease. Finally, 495.000 deaths worldwide are due to asthma every year [3]. Asthma continues to be a major source of global burden in terms of both social and economic (including direct and indirect costs) aspects. Asthma leads to high direct medical costs (hospital admissions, physician visits, and medications) and indirect costs due to its impact on productivity loss and premature death [1,4].

The strongest risk factors for developing asthma are inhaled substances and particles that may provoke allergic reactions or irritate the airways, in which chemical irritants in the workplace are a typical example [1]. When a substance or condition at work causes asthma, it is called work-related asthma. Work-related asthma (WRA) comprises two major entities, occupational asthma (OA-, asthma induced by sensitizer or irritant work exposures) and work-exacerbated asthma (WEA-pre-existing or concurrent asthma worsened, but not caused, by work factors: aeroallergens, irritants, or exercise…) [5,6]. In the textile industry, work-related asthma is considerable and several agents such as cotton dust and dyes may cause this condition [7]. The prevalence of work-related asthma among textile workers has been studied in several countries: 9.1% in Thailand [8], 0.9% of women and 1.1% of male textile workers in Croatia [9], 4.0% and 5.0% in Pakistan in 2013 and 2015 respectively [10,11]. Meanwhile, work-exacerbated asthma is more common with about 25% to 50% of working adults with known asthma causing exacerbations with asthma symptoms related to their workplace [12]. These symptoms include wheezing, cough, chest tightness, and/or shortness of breath.

Vietnam has a developed textile industry with more than 3800 companies, more than 2 million workers, and ranks fifth in the world in textile and apparel exports [13]. The textile working environment has a lot of factors (such as cotton dust, noise, humidity, lack of light…) that contribute negatively to the health of workers despite the regulations on annually measuring the working environment and periodic medical examinations for the workers [13]. Indeed, in Vietnam, well-being at work is an issue of increasing concern. Currently, the list of occupational diseases covered by insurance has expanded to 34 [14], including occupational asthma, which is a major occupational problem in the textile industry.

Nam Dinh province located in the center of the Southern Red River Delta region of Vietnam is called “The Textile City” of Viet Nam, with about 480 companies and more than 200,000 active workers, accounting for 8% of textile companies in the country [15]. Currently, besides traditional businesses such as Nam Dinh Textile Factory, Song Hong Garment Joint Stock Company…, the whole province has two operating industrial parks, including Bao Minh and Hoa Xa, attracting many textile enterprises, a large-scale garment at home and abroad, creating jobs for many workers. In addition, the province has 24 district-level industrial clusters established, of which 19 industrial clusters with a total area of 352.5 ha have been put into operation, providing space demand for the first textile enterprises production development and business investment. There is a school specializing in training high-tech workers for the textile industry, Nam Dinh Textile Industry College, that uses modern high-tech equipment compared to textile vocational training schools in Vietnam. In recent years, there have been many studies assessing the working environment and the health status of textile workers [13,15]. However, in the last fifteen years, there have not been any studies to assess asthma disease and asthma-related allergies to cotton dust among textile workers. Thus, there is a need to understand the prevalence of cotton dust-related allergy asthma, and other respiratory conditions as well as relevant health hazards in this industry to strengthen the existing system to protect workers’ health. The object of this study is to determine the prevalence of cotton dust-related allergic asthma and associated factors among textile workers in Nam Dinh province, Vietnam.

## 2. Materials and Methods

### 2.1. Study Design

A descriptive cross-sectional survey was performed between May and November 2016 to ascertain the cotton dust-related allergic asthma and some potential explanatory variables among workers in two textile companies (Nam Dinh Yarn and Song Hong Garment) of Nam Dinh, Vietnam.

### 2.2. Eligibility Criteria

We selected all workers (1125) who participated in all stages of the production process (harvesting, preparatory processes, spinning, weaving, and finishing) of 2 companies and meet the following criteria: − Directly participate in the production process,− Have worked in their current position for at least 12 months, and− Agree to participate in the study.

A total of 1082/1125 (368 in Nam Dinh Yarn and 714 in Song Hong Garment) workers participated in this study.

### 2.3. Criteria for Making the Diagnosis of Asthma and Cotton Dust-Related Allergic Asthma

Diagnosis criteria for asthma are based on GINA (Global Initiative For Asthma)’s guidelines updated in 2016 with 2 key defining features: a history of respiratory symptoms such as wheeze, shortness of breath, chest tightness, and cough that vary over time and in intensity, AND variable expiratory airflow limitation [16].

Criteria for determining allergic asthma associated with cotton dust in the workplace [17]:− have been diagnosed with asthma according to the criteria above,− have a family or personal history of allergy,− serum IgE > 100 UI/mL, AND− positive for skin test (Prick test) with cotton dust.

### 2.4. Data Collection Instruments

We used the following tools to collect data:− A sociodemographic questionnaire was used to collect general information, such as age, gender, height, weight, marital status, smoking status, personal history of asthma across their whole life, family history of allergy, and work-related information, including seniority, working position, and time of cotton dust exposure per day.− An asthma screening questionnaire was developed based on GINA guidelines for the diagnosis of asthma.− Respiratory function measurement tool: CHEST HII 801 spirometer (2012, Japan) and Ventolin Inhaler 100mcg for bronchodilator reversibility test.− Materials to serve the prick test [16] with cotton dust allergen.− Materials for blood collection to quantify serum IgE concentration.

### 2.5. Research Progress

This study went through the following stages:

Step 1: We worked with the trade unions of 2 companies to build the research plan, organizational processes and provided an information sheet to all their 1125 workers before the study. The survey was set within the workers’ annual medical checkup sessions of the companies. In total, 1082 workers agreed to participate in the study, signed the consent forms, and were then interviewed about socio-demographic information, occupational characteristics, and asthma screening (symptoms, history of asthma, history of allergy). Parallel to the interview, all participants were examined for allergic rhinitis and allergic sinusitis performed by the otorhinolaryngology specialists of the National Otorhinolaryngology Hospital. There were 129 workers with suspected asthma symptoms. The total timing of this step took place in 3 weeks with 8 trained interviewers per day.

Step 2: The respiratory function was measured in those with symptoms and a history of suspected asthma. The following parameters were measured: − vital capacity (VC)− forced vital capacity (FVC)− forced expiratory volume in the first second (FEV1)− Gaensler index is calculated by the following formula: Gaensler = FVC/FEV1.

A person exhibits obstructive ventilatory disorders when: FEV1 <80% and Gaensler index <75% of the theoretical value.

People with obstructive ventilation were given a bronchodilator reversibility test with Salbutamol (Ventolin Inhaler 100 mcg). This test is considered positive when the FEV1 at the second measurement (after Salbutamol spray) increases ≥ 12% more compared to the first test and/or the absolute value of the FEV1 is greater than 200 mL. A total of 80 workers had obstructive ventilation and were positive at the bronchodilator reversibility test (i.e., diagnosed with asthma).

Step 3: People with diagnosed asthma had a prick test with cotton dust allergen and a determination of IgE concentration for the purpose of diagnosing allergic asthma in association with cotton dust. Forty-six workers were found to have allergic asthma in association with cotton dust.

The study processes are presented in Figure 1.

### 2.6. Statistical Analysis

SPSS version 22.0 software (SPSS Inc. IBM, Armonk, NY, USA) was used for data analysis. Quantitative variables were described as mean/median and standard deviation. Prevalence (number, percentage, 95% confidence interval) of cotton dust-related allergic asthma in each category according to sociodemographic, occupational, and allergic rhinitis and sinusitis variables was calculated. A chi-square test (or Fisher’s exact test) for qualitative variables was used to compare characteristics of the sample according to cotton dust-related allergic asthma. Multiple logistic regression was used to study the associated factors of cotton dust-related allergic asthma. A stepwise backward selection strategy was applied along with multivariate logistic regression to have reduced models. This strategy used a *p*-values threshold of <0.2 at a log-likelihood ratio test for predictors included. The level of significance was set at a *p*-value of less than 0.05.

### 2.7. Ethics

All study procedures involving human subjects were approved in advance by the Institutional Review Boards at the Vietnam National Institute of Occupational and Environmental Health. Data collection procedures also were approved by the directors of the two companies. Written informed consent was obtained from all participants.

## 3. Results

### 3.1. Characteristics of The Study Population

Among the workers participating in the study, 69.1% were female (748/1082). The mean age was 34.8 years (SD = 8.6), half of them were between the ages of 30 and 39 (49.3%) and half had a working experience of more than 10 years (50.3%). Most of them were married (96.65%) and non-smokers (98.3%). Percentage of asthma history and family history of allergic reaction were 0.6% and 5.9% respectively, while 4.3% had allergic sinusitis and more than half (52.1%) had allergic rhinitis. Fifty-six dot 9 percent of workers were exposed to cotton dust from 5 to 8 h in 1 working day and 25.3% from 9 to 11 h. The percentage of non-exposed to cotton dust in the working environment was 12.9% (Table 1).

### 3.2. Prevalence of Cotton Dust-Related Allergic Asthma

In a total of 1082 workers, 129 (11.9%) had suspected asthma symptoms, 80 (7.4%) were diagnosed with asthma, and 46 (4.3%) were diagnosed with cotton dust-related allergic asthma.

The prevalence of cotton dust-related allergic asthma was 3.6% in men and 4.5% in women (*p* = 0.473). This prevalence tended to increase with age (*p* = 0.152), seniority (*p* = 0.001) and time of cotton dust exposure per day (*p* = 0.004). People marked overweight had a higher prevalence than non-overweight people (14.6% versus 3.8%, *p* < 0.01). Finally, workers with a history of asthma, a family history of allergic and a diagnosed allergic rhinitis had a statistically significantly higher prevalence of cotton dust-related allergic asthma than workers without history (all *p*-values less than 0.01) (Table 1).

### 3.3. Associated Factors with Cotton Dust-Related Allergic Asthma among Textile Workers

A *p*-value of less than 0.2. univariate was used in analyses first used to find the factors that are potentially associated with cotton dust-related allergic asthma. These factors were put into the multivariate model of logistic regression. Final model results showed that overweight, seniority of more than 10 years, history of asthma, allergic rhinitis, family history of allergy, and duration of exposure to cotton dust per day were statistically significantly associated with cotton dust-related allergic asthma.

In detail, overweight had OR = 5.11, *p* = 0.002 (reference category was non-overweight); compared to a seniority under 5 years, workers with a seniority over 10 years had the highest prevalence of cotton dust-allergic asthma with an OR = 4.12 and *p* = 0.012; people with current allergic rhinitis or a history of asthma, a family history of allergy had a higher prevalence of cotton dust-related allergic asthma compared to those without allergy, with an OR and *p* values respectively at 3.67, 0.011; 8.11, 0.009 and 9.64, <0.001; workers exposed to cotton dust for 1–8 h (OR = 4.52, *p* = 0.029) and 9 to 11 h (OR = 2.91, *p* = 0.004) per day had a higher prevalence of cotton dust-related allergic asthma compared with workers not exposed to cotton dust (Table 2).

## 4. Discussion

The prevalence of work-related asthma is poorly defined in Vietnam. To the best of our knowledge, this study is the first study estimating asthma prevalence among textile workers using four different criteria i.e., physician-diagnosed (according to GINA 2016), respiratory function assessment, skin prick test with cotton dust allergen, and quantification of serum IgE concentration. Furthermore, this study was carried out with a large number of participants (1082 textile workers). These strengthened its capacity to provide meaningful information to better understand the prevalence and associated factors of cotton dust-allergic asthma in Vietnam.

In this study, we found 129 (11.9%) workers with suspected asthma symptoms. Among them, eighty (7.4%) were diagnosed with asthma. More than half of all asthma cases (46/80) were associated with cotton dust allergies (3.6% in men and 4.5% in women) following a lung function assessment, skin prick test, and serum IgE quantification. This prevalence of asthma is higher than what is found in the general population in Vietnam. For example, one research conducted in Hanoi in 2010 among 7008 adults aged 21 to 70 years old using a self-reported questionnaire showed that the prevalence of all-cause asthma was 5.6% [18]. Another study performed in Dalat using a questionnaire plus a lung function assessment mentioned that the prevalence of asthma was 2% [19]. One notes that all the textile workers are required to undergo a medical examination before being employed, to be sure that they are all healthy and able to perform textile works. Our results show that the working environment might have an important influence on the asthma rate of workers.

However, the results found in our study are similar to those of several studies conducted in other countries. A cross-sectional study of Nafees et al. in 2013 among 372 adult male textile workers from the spinning and weaving sections of 15 textile mills in Karachi, Pakistan indicated a low prevalence of asthma (4%) out of 8% having obstructive disorders [10]. In Tunisia, the prevalence of occupational asthma among textile workers was 8% according to research studies compiled by Chaari et al. in 2011 [7], and 4% in another study by the same author on 600 apprentices in a textile and clothing vocational training center in the Monastir area [20]. Likewise, in the Czech Republic population, the rate of occupational asthma and occupational allergic rhinitis of total asthma and rhinitis incidence has been shown to fluctuate between 5–15% [21]. Nevertheless, the results of the study in Pakistan showed that the asthma prevalence in the textile workers was much lower than that one found in the general population (up to 20%), possibly due to some protective effect of endotoxin exposure [10]. This suggests that there is a need to implement specific measures of prevention, screening and care for dust-related asthma given its high prevalence among workers in the textile industry.

To take into account factors that may be related to cotton dust-related allergic asthma, a multivariate regression model was conducted after eliminating the variables individually associated with asthma with *p*-values greater than 0.2. The results showed that overweight, seniority more than 10 years, history of asthma, allergic rhinitis, family history of allergy, and exposure to cotton dust from 1 to 11 h in the working environment were relevant factors compared to the reference categories. Two other factors (working position and smoking) were significantly associated with allergic asthma in bivariate analyses but these factors were not significantly associated in multivariate analyses. These findings were reported for the first time in Vietnam but have also been recorded in many other studies around the world. Many studies have provided evidence of an association between overweight/obesity and respiratory disorders, including asthma. Although the understanding of the mechanisms of asthma in obese patients is still unclear, most studies conclude that being overweight and having asthma are associated. A study conducted in the United State of America based on 3 national time trends studies on obesity among adults with asthma indicated that the prevalence of obesity was higher in the asthma group (21.3–32.8%) compared with the non-asthma group (14.6–22.8%) [22]. A retrospective study of 143 adults found a similar association between obesity prevalence and asthma severity [23]. Based on the evidence on this association, and evidence of chronic inflammatory response linking overweight/obesity and asthma will help to address overweight/obesity issues in asthma control [24]. Regarding seniority, workers who worked more than 10 years had a higher risk of asthma than those who worked less than 5 years. This result is similar to the results of some other studies. The Pakistani study of 327 workers showed that the duration of work ≥10 years in the textile industry was an important predictor of asthma [10].

Of the factors that cause or are related to occupational asthma, especially in the textile industry, dust is an important allergen, through its content of latex particles, cotton, pollens, dyes, solids suspended in air… [7,25]. Cotton dust among textile workers, both in terms of time and concentration exposure, leads to impaired lung function. A study by Ali et al. set among 303 adult male textile workers has shown that every mg/m^3^ increase in dust concentration was associated with a 5.4% decline in FEV1 (forced expiratory volume in the first second) and therefore may be associated with asthma [26]. Another study by Hinson et al. in 2016 aimed to evaluate the respiratory medical among the textile workers exposed to cotton dust involved 656 subjects exposed to cotton dust and 113 non-exposed subjects in a Beninese cotton industry company. The main results revealed that subjects exposed to cotton dust have more respiratory symptoms than unexposed subjects (36.9% vs. 21.2%); and the prevalence of chronic cough, expectorations, dyspnea, asthma, and chronic bronchitis are higher among the exposed in comparison with unexposed subjects [27].

Personal history and family history of asthma, allergic asthma, and allergies increase the probability that the respiratory symptoms are due to asthma [2]. A study by Darika Wortong et al. showed that the highest risk factors of asthma were a family history of asthma and a history of atopy [28]. Among working adolescents, Erkan Cakir et al. found that a family history of allergy, history of allergic rhinitis, and active smoking were risk factors for asthma and related symptoms [29].

Allergic rhinitis subjects have been found to have a 3-fold increased risk of asthma compared to subjects without allergic rhinitis (23.8 and 7.5%, respectively) [30]. A study by Dorothée Provost et al. among the French working population revealed that 72.5% of current asthmatics had allergic rhinitis [31]. In a review, Mike Thomas stated that allergic rhinitis is very common in patients with asthma, with a reported prevalence of up to 100%. His article also pointed out that comorbid allergic rhinitis is a marker for more difficulty in controlling asthma and with worse asthma outcomes [32]. Allergic rhinitis is not only associated with the occurrence and symptoms of allergic asthma, but also a factor that influences the management and treatment of asthma.

Our study shows that workplace exposure to cotton dust increases the likelihood of allergic asthma. However, because this is a cross-sectional study, it is not possible to accurately assess whether the duration of exposure increases the risk of allergic asthma. A study by Boubopoulos et al. on 443 cotton workers concluded that despite the reduction in cotton dust concentration, asthma, respiratory symptoms, and pathology are the most common findings in cotton workers and that it depends on the duration of exposure whether workers are smokers or not and whatever the nature of the cotton dust [33]. Therefore, in order to reduce respiratory diseases for cotton workers, it is necessary to comprehensively intervene on many factors and to focus on the groups of high seniority and family allergic antecedent workers.

### Weaknesses and Strengths of the Study

One of the strengths of this study is that it is one of the first studies on asthma among cotton workers not only in Vietnam but also in Southeast Asia. The study had a large sample size for a cross-sectional description with a variety of demographic characteristics. In addition, the study used many standard tools to serve the diagnosis of allergic rhinitis, allergic sinusitis, and atopic asthma according to the guidance of GINA. This is a good reference for researchers in the field of occupational health and preventive medicine. It is also an important source of data for managers of textile companies to improve the health of their employees.

Besides the above-mentioned strengths, the study has weaknesses that need to be considered in further research work on the topic. Due to the cross-sectional study design, causal relationships between relevant factors and allergic asthma have not been established. Sampling was only conducted at 2 textile factories in Nam Dinh province; this may make the sample not representative of all cotton workers in the province in particular as well as in Vietnam in general. Calculation of the minimum sample size for a cross-sectional study should also be performed. Further, recall bias might be also encountered during direct interviews with study participants. Yet, the interviewers were well-trained, and the questionnaire was pre-tested before being used in the study. This has probably mitigated the study’s weaknesses. This study also has not mentioned other allergen factors, other types of dust, and the nature of cotton dust such as its concentration.

This study used the GINA criteria for asthma diagnostics, which are community diagnostic standards, not exclusive to occupational asthma. In addition, due to resource constraints, repeated measurements of respiratory function to assess changes in FEV1 at different times of the day have not been performed, which has somewhat reduced the quality of the diagnosis. However, when referring to similar studies in the world, we found that spirometry’s role is primarily used for screening, not necessarily for giving a clinical judgment.

The above limitations are also the foundation for recommendations for further studies on allergic asthma in order to obtain consistent results and provide more useful results for clinicians and public health professionals.

## 5. Conclusions

This study has revealed a high prevalence of allergic asthma and cotton dust-related allergic asthma among textile workers in Nam Dinh, Vietnam compared to the general population, and has also pointed out several factors that are potentially associated with the last condition. The working environment is probably playing an important role in the occurrence of asthma in this population. Prevention and care of dust-related allergic asthma among workers in the textile industry in Vietnam is needed. Future in-depth studies with more relevant designs are necessary to supplement the limitations of this study.

## Figures and Tables

**Figure 1 ijerph-18-09813-f001:**
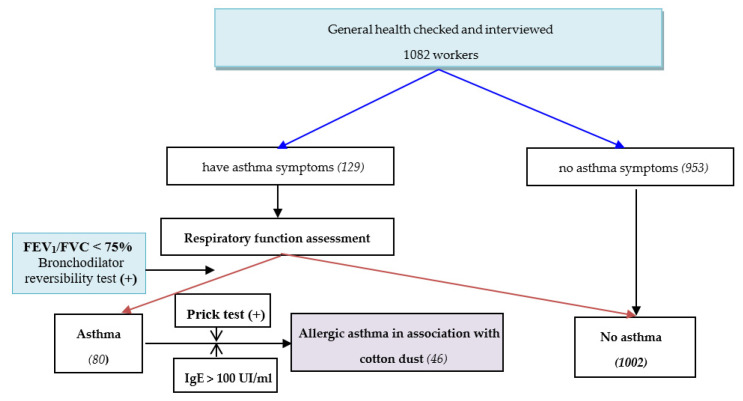
The study processes. (+): positive.

**Table 1 ijerph-18-09813-t001:** Demographic and cotton dust-related allergic asthma characteristics among workers.

Variables	*N* = 1082	*n* (%)	Cotton Dust-Related Allergic Asthma				*p*-Value
			Yes		No		
			n	%	*n*	%	
Gender	Male	334 (30.9)	12	3.6	322	96.4	0.473
	Female	748 (69.1)	34	4.5	714	95.5	
Age group (years)	20–29	221 (20.4)	5	2.3	216	97.7	0.152
	30–39	533 (49.3)	21	3.6	512	96.1	
	40–49	215 (19.9)	14	6.5	201	93.5	
	≥50	113 (10.4)	6	5.3	107	94.7	
Overweight **	Yes (BMI ≥ 25)	48 (4.4)	7	14.6	41	85.4	<0.01
	No	1034 (95.6)	39	3.8	995	96.2	
Seniority	≤5 years	242 (22.4)	4	1.7	238	98.3	0.001
	6–10 years	296 (27.3)	6	2.0	290	98.0	
	>10 years	544 (50.3)	36	6.6	508	93.4	
Working position (location of company)	Yarn Factory	368 (34.0)	22	6.0	346	94.0	0.043
	Garment comp.	714 (66.0)	24	6.4	690	93.6	
Marital status	Others	37 (3.4)	1	2.7	36	97.3	0.527 *
	married	1045 (96.6)	45	4.3	1000	95.7	
Smoking	smoke	38 (1.7)	5	13.2	33	86.8	0.006
	Non-smokers	1044 (98.3)	41	3.9	1003	95.1	
History of asthma	Yes	16 (1.5)	4	25.0	12	75.0	0.004 *
	No	1076 (98.5)	42	3.9	1024	96.1	
Family history of allergy	yes	191 (17.7)	32	16.8	159	83.2	<0.001
	No	891 (82.3)	14	1.6	877	98.4	
Allergic rhinitis	yes	564 (52.1)	41	7.3	523	92.7	<0.01
	no	518 (47.9)	5	1.0	513	99.0	
Allergic sinusitis	yes	46 (4.3)	4	8.7	42	91.3	0.127 *
	No	1036 (95.7)	42	4.1	994	95.9	
Time of cotton dust exposure per day	no	140 (12.9)	2	1.4	138	98.6	0.004
	1–4 h	52 (4.8)	1	1.9	51	98.1	
	5–8 h	616 (56.9)	23	3.7	593	96.3	
	9–11 h	274 (25.3)	20	7.3	254	92.7	

*: Fisher’s Exact Test. **: According to the IDI and WPRO classification.

**Table 2 ijerph-18-09813-t002:** Factors associated with allergic asthma among textile workers.

Independent Variables		Crude OR ^a^ OR (95%CI)	Adjusted OR ^b^ OR (95%CI)	*p*-Value
Overweight	No (reference)	-	-	0.002
	Yes	4.35 (1.84–10.32)	5.15 (1.86–14.28)	
Seniority	≤5 years (reference)	-	-	
	6–10 years	1.05 (0.35–3.16)	1.32 (0.57–3.95)	0.520 0.012
	>10 years	2.54 (1.05–6.14)	4.12 (1.37–12.38)	
Working position (location of company)	Garment company (reference)	-	-	0.297
	Yarn Factory	1.83 (1.01–3.31)	1.45 (0.72–2.93)	
History of asthma	No (reference)	-	-	0.009
	Yes	8.13 (5.52–26.26)	8.11 (1.71–18.55)	
Allergic rhinitis	No (reference)	-	-	0.011
	Yes	8.04 (3.15–20.52)	3.67 (1.34–10.03)	
Family history of alergy	No (reference)			<0.001
	Yes	12.61 (6.58–24.16)	9.64 (4.73–19.64)	
Time of cotton dust exposure per day	No (reference)	-	-	
	1–8 h	2.57 (0.6–11.0)	4.52 (1.17–17.49)	0.029
	9–11 h	5.43 (1.25–23.59)	2.91 (1.41–5.99)	0.004
Smoking	Non (reference)	-		0.199
	Smoke	12.61 (6.57–24.16)	2.15 (0.67–6.91)	

Abbreviation: OR, Odds Ratio; CI, Confidence Interval; ^a^ Univariate analysis; ^b^ Multiple logistic regression.

## Data Availability

The EXCEL/SPSS data used to support the findings of this study are available from the corresponding author upon request.

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
