# Peer review of "The Cotton Dust-Related Allergic Asthma: Prevalence and Associated Factors among Textile Workers in Nam Dinh Province, Vietnam"

_ijerph, 2021, doi:10.3390/ijerph18189813_

Round 1
Reviewer 1 Report
Dear Authors,
Thanks for the opportunity to read your manuscript. This is an interesting study and an important one from the perspective of occupational health and safety in a labour intensive industry. Hence, I felt the discussion could be extended on the significance of the findings on disease prevention, i.e., to reduce occupation related asthma due to cotton dust in Vietnam, and - if you have the evidence - what are the opportunities and challenges of achieving this in the Vietnam context.
Specifically, there are some places where you may want to consider revisiting or revision:
- In page 2 you listed international study results for the prevalence of work-related asthma and said it was limited and controversial - why are these results limited and controversial?
- This sentence may need a second look: it is slightly disconnected between the two parts. "In Vietnam, although the occupational health sector is still underdeveloped, occupational diseases and their prevention are increasingly concerned." What is 'occupational sector'? Why are occupational diseases are increasingly concerned? Maybe some figures on the increase of work-related diseases can show why this is worrying.
- In the final conclusion, it may be good to update what is the latest government measures and/or employers' responses trying to control or prevent work-related asthma in the textile industry in Vietnam. Without this information, it feels like as if this problem has been left out there for workers themselves to handle, despite the recommended solutions (mentioned in your paper).
- How did the research team interview all 1082 workers? This must be a very intensive fieldwork task.
- Could you discuss the potential bias occurred in cross-sectional data collection and what future actions might be taken to offset this weakness?
- I wondered if you could explain this - your result (7.4%) is said to be similar to the findings from other countries, but those figures are ranging from 2% to more than 11%. As the manuscript says, this is within 20% of the general population. Is this why these results are similar?
- The discussion of your research results is somehow overwhelmed by comparative data, not what your own data and findings mean to the subject. It is essential to compare but the paper should really focus more on what these results tell us, not what other results form other studies tell us.
- Related, maybe you (in the discussion section) can consider to further discuss why the study is important/significant, what aspects of the findings are uniquely different, and what are the theoretical and practical implications. For example, the findings about allergic rhinitis, overweight/obesity and age are all found to be factors leading to work-related asthma, and your discussion seems to suggest these are in line with previous findings elsewhere. So, the conclusion ("it is necessary to comprehensively intervene on many factors") appears to be rather generic. Given the country specific environment in terms of labour market and the rapid growth of textile industry, there must be something unique to the factors distinguished from other countries. This can help you to highlight why some of the factors stand out and hence contributing either literature on occupational health or practice for disease prevention.
- There are some minor grammar errors throughout. Please proofread again to minimize this problem.
Finally, good luck to your paper.
Author Response
Dear Reviewer,
Thank you very much for your comments that helped us to improve the quality of our report and manuscript. We tried to address all issues they raised, and you will find below responses on all points. Hope this answer to all your questions and suggestions:
- In page 2 you listed international study results for the prevalence of work-related asthma and said it was limited and controversial - why are these results limited and controversial?
Answer: we have modified this sentence, in line 56-59 of the manuscript.
- This sentence may need a second look: it is slightly disconnected between the two parts. "In Vietnam, although the occupational health sector is still underdeveloped, occupational diseases and their prevention are increasingly concerned." What is 'occupational sector'? Why are occupational diseases are increasingly concerned? Maybe some figures on the increase of work-related diseases can show why this is worrying.
Answer: we have modified this sentence, in line 62-65 of the manuscript.
- In the final conclusion, it may be good to update what is the latest government measures and/or employers' responses trying to control or prevent work-related asthma in the textile industry in Vietnam. Without this information, it feels like as if this problem has been left out there for workers themselves to handle, despite the recommended solutions (mentioned in your paper).
Answer: we added some more information about the government measures to control or prevent work-related asthma in the textile industry in Vietnam, in the line 63-64 of the manuscript.
- How did the research team interview all 1082 workers? This must be a very intensive fieldwork task.
Answer: we added more information about organization of the study, cited in the line 133-135 of the manuscript.
- Could you discuss the potential bias occurred in cross-sectional data collection and what future actions might be taken to offset this weakness?
Answer: we have added this in line 331-334 of the manuscript.
- I wondered if you could explain this - your result (7.4%) is said to be similar to the findings from other countries, but those figures are ranging from 2% to more than 11%. As the manuscript says, this is within 20% of the general population. Is this why these results are similar?
Answer: we have modified this sentences on the manuscript in adding some more data of asthma prevalence in the general population, in the line 228-238 of the manuscript.
- The discussion of your research results is somehow overwhelmed by comparative data, not what your own data and findings mean to the subject. It is essential to compare but the paper should really focus more on what these results tell us, not what other results from other studies tell us.
Answer: we have updated the discussion section in order to improve this section in different aspects including suggestions from reviewers and especially highlighting the meaning of what was found in the results. Hope this is suitable for the results we have from our study.
- Related, maybe you (in the discussion section) can consider to further discuss why the study is important/significant, what aspects of the findings are uniquely different, and what are the theoretical and practical implications. For example, the findings about allergic rhinitis, overweight/obesity and age are all found to be factors leading to work-related asthma, and your discussion seems to suggest these are in line with previous findings elsewhere. So, the conclusion ("it is necessary to comprehensively intervene on many factors") appears to be rather generic. Given the country specific environment in terms of labour market and the rapid growth of textile industry, there must be something unique to the factors distinguished from other countries. This can help you to highlight why some of the factors stand out and hence contributing either literature on occupational health or practice for disease prevention.
Answer: we tried to make clearly and discuss of our findings in the manuscript following the reviewers suggestion, please see in the manuscript (line 216-259)
- There are some minor grammar errors throughout. Please proofread again to minimize this problem.
Answer: we have been proofreading carefully whole manuscript to minimize the grammar errors and correct different sentences with help of a native English-speaking colleague.
Reviewer 2 Report
The manuscript entitled “The cotton dust-related allergic asthma: Prevalence and associated factors among textile workers in Nam Dinh province, Vietnam” aimed to measure the prevalence and associated factors of cotton dust-related allergic asthma among textile workers in Nam Dinh Province, Vietnam. In addition, work of this paper is practical and logical. However, there are some problems to be further improved as well:
- The research method of this paper is not innovative enough. In fact, authors need to highlight their own innovative contributions.
- In the process of data analysis in this paper, the author only considers single-factor variables, and the scope is too narrow. Multiple factors can be considered as variables at the same time.
- The abstract is too cumbersome and not clear enough, so it needs further improvement.
- The conclusion is too simple, without discussion and exploration of the significance of this study.
Author Response
Dear Reviewer,
Thank you very much for your comments that helped us to improve the quality of our report and manuscript. We tried to address all issues they raised, and you will find below responses on all points. Hope this answer to all your questions and suggestions:
- The research method of this paper is not innovative enough. In fact, authors need to highlight their own innovative contributions.
Answer: we have modified the discussion part, in adding the innovative points of our study, cited in line 216-225, and another part in the weakness and strengths of the study.
- In the process of data analysis in this paper, the author only considers single-factor variables, and the scope is too narrow. Multiple factors can be considered as variables at the same time.
Answer: Before the study, we have looked carefully all potential factors in other studies. With your suggestions, we have revised back again and considered all the potential factors mentioned in the literature and found some significative factors in this manuscript. We did not find any multiple factors variable being reported significative in our analysis. So further study is needs for this.
- The abstract is too cumbersome and not clear enough, so it needs further improvement.
Answer: we have modified this part. Please see in the manuscript.
- The conclusion is too simple, without discussion and exploration of the significance of this study.
Answer: we have modified this part. Please see in the manuscript.
Round 2
Reviewer 2 Report
The authors have made a commendable effort to address most of reviewers’ concerns within the available space. I am satisfied with the revised version of the paper. I recommend published in this journal.